# Omnidirectional and Broadband Antireflection Effect with Tapered Silicon Nanostructures Fabricated with Low-Cost and Large-Area Capable Nanosphere Lithography

**DOI:** 10.3390/mi12020119

**Published:** 2021-01-23

**Authors:** Sangho Kim, Gwan Seung Jeong, Na Yeon Park, Jea-Young Choi

**Affiliations:** 1Department of Materials Science & Engineering, Dong-A University, Busan 604-714, Korea; vohno@nate.com; 2Department of Metallurgical Engineering, Dong-A University, Busan 604-714, Korea; wjdrhkstmd12@gmail.com (G.S.J.); nayeon2385@gmail.com (N.Y.P.)

**Keywords:** nanostructure, gradient refractive index change, antireflection, spin-coating, solar cell

## Abstract

In this report, we present a process for the fabrication and tapering of a silicon (Si) nanopillar (NP) array on a large Si surface area wafer (2-inch diameter) to provide enhanced light harvesting for Si solar cell application. From our *N*,*N*-dimethyl-formamide (DMF) solvent-controlled spin-coating method, silica nanosphere (SNS in 310 nm diameter) coating on the Si surface was demonstrated successfully with improved monolayer coverage (>95%) and uniformity. After combining this method with a reactive ion etching (RIE) technique, a high-density Si NP array was produced, and we revealed that controlled tapering of Si NPs could be achieved after introducing a two-step RIE process using (1) CHF_3_/Ar gases for SNS selective etching over Si and (2) Cl_2_ gas for Si vertical etching. From our experimental and computational study, we show that an effectively tapered Si NP (i.e., an Si nanotip (NT)) structure could offer a highly effective omnidirectional and broadband antireflection effect for high-efficiency Si solar cell application.

## 1. Introduction

The fabrication of silicon (Si) surface nanostructures has gained a great deal of attention in the past several decades because Si surface nanostructures have great potential due to their enhanced optical and electrical advantages that can differentiate them from bulk Si materials. In particular, for photovoltaic (PV) applications there is growing interest in nanostructures because of their excellent antireflection (AR) effect in the broadband wavelength range, which is realized by a gradient refractive index change in the nanostructured layer [1,2,3,4,5,6]. Additionally, surface reflection with the nanostructured layer has much less dependency on the incident angle of light, offering an omnidirectional AR effect that is highly desired for high-efficiency solar cell applications [7]. Therefore, surface nanostructure fabrication offers substantial advantages compared with conventional geometric structures such as alkaline solution (i.e., KOH) etched pyramid structures (10~20 µm scale), which cannot offer effective reduction of broadband surface reflection (more than 10% reflection) [8]. Consequently, an additional AR coating (ARC) must be applied by introducing dielectric layers of different refractive indices, such as silicon nitride or silicon dioxide [9,10,11]. Furthermore, the degree of reflectance suppression with these dielectric layers is not consistent, as it depends on the incident light angle and spectral range [12].

The extensive recent research on nano-scale (or sub-micron) surface structures has revealed that optimized nanostructure periods, scales, and shapes can deliver greatly enhanced light absorption, even with a significant reduction in the thickness of the active layer. This can be attributed to elevated light interference and diffraction efficiency in the active layers, providing improved light-trapping in the absorption layer [13,14,15]. A report by Han et al. showed that an optimized Si-nanostructure-enabled ultra-thin Si solar cell could provide improved absorption close to the Lambertian limit, thus producing a comparable photocurrent to a conventional thick Si solar cell [16]. Because of the great potential of nanostructures for PV application, intensive investigations have been made by numerous research groups to develop various shapes and scales of nanostructures [17,18]. However, many of the current nano-fabrication techniques, such as e-beam [19,20] or laser interference techniques [21,22], have been limited to small-area applications because of their extremely time-consuming and expensive processes. These have been barriers for practical and industrial applications, but reactive ion etching (RIE) has recently been widely investigated in an attempt to overcome these barriers arising from conventional techniques. Unlike conventional approaches, RIE is a fast and cost-effective etching technique, even on large surface areas, and provides phenomenal control of etching orientation and selectivity by using appropriate etching chemistries. However, for desired shapes and scales of nanostructure fabrication with RIE, a time-consuming and high-cost photolithography (PL) process for nano-patterning must be introduced, and this makes the RIE nano-fabrication process less attractive for PV applications.

In this study, we describe a simple large-area-capable (≥2-inch diameter Si wafer) Si nanostructure fabrication process that enables the processing of Si nanopillars (NPs) with desired shape on an expanded Si surface area. The surface nano-lithography was achieved with a silica nanosphere (SNS) lithography technique in place of the conventional high-cost and time-consuming PL process. For uniform and large-area SNS monolayer coating, we used our *N*,*N*-dimethyl-formamide (DMF) solvent-controlled SNS spin-coating method, whose superior effectiveness and simplicity was confirmed in our previous report [17]. With an SNS spin-coated Si surface, a two-step RIE process with (1) a CHF_3_/Ar gas step for SNS (or SiO_2_) selective etching over Si and (2) a Cl_2_ gas step for Si vertical etching was used to fabricate Si NPs with different top diameters. In addition, computational optical modeling was also performed to demonstrate the incident-angle-dependent AR effect for fabricated Si NPs. From these experimental and computational results, we successfully reveal that a well-tapered Si NP (i.e., an Si nanotip (NT)) structure can provide a highly effective omnidirectional and broadband AR effect for high-efficiency Si solar cell application.

## 2. Experimental Section

Silicon wafers of polished 2-inch diameter were used as substrates. Dry silica nanospheres (SNSs) of 310 nm were obtained from Bangs Laboratories, Inc. The common solvents (acetone and methanol) used in the experiment were purchased from Chemical Strategies, Inc. H_2_O_2_ (30 wt% in water) was purchased from Honeywell, H_2_SO_4_ (purity 96.0%) from KMG Chemicals, Inc., and *N*,*N*-dimethyl-formamide (DMF, purity > 99%) from Sigma-Aldrich.

Prior to SNS spin-coating, the Si substrates were cleaned with common solvent cleaning with acetone, methanol, and DI water and submerged in a piranha solution (H_2_O_2_:H_2_SO_4_ = 1:4 *v*/*v*) for 15 min to remove organic residues and improve the hydrophilicity of the Si surface. The cleaned Si substrates were then rinsed with DI water and dried by a spin rinse dryer at 1000 rpm for 10 min. Next, the substrates were placed on a spin-coater and exposed to the air under normal ambient laboratory conditions (temperature: 20~25 °C, humidity: 25~35%). The spin-coater was programmed with an acceleration speed of 100 rpm/s, a maximum speed of 2000 rpm, and a duration of 120 s. Of the SNS solution (15 wt% of SNS in DMF), 300 μL was dropped on the surface for spin-coating.

SNS-deposited silicon substrates were transferred to the chamber of a reactive ion etcher (PlasmaLab 80+). The RIE process was set up as a two-step process of (1) CHF_3_/Ar gas etching for size reduction of the SNSs and (2) Cl_2_ gas etching for anisotropic or semi-anisotropic etching of the Si [22]. The etching conditions used were a 25 sccm (CHF_3_)/25 sccm (Ar) gas flow rate, 100 W RF power, and 30 mTorr chamber pressure for CHF_3_/Ar etching and a 10 sccm gas flow rate, 100 W RF power, and 30 mTorr chamber pressure for Cl_2_ etching.

Scanning electron microscopy (FEI XL-30) was used to observe the Si surface, and the coverage of the SNS monolayer was calculated by direct counting of the SNS-covered area through ImageJ image analysis software (National Institutes of Health, Bethesda, MD, USA) [23], which has been widely used for particle-related surface analysis [24,25,26,27,28]. The wettability of solvent on the Si surface was determined by contact angle measurement with an EasyDrop contact angle measurement system (KRUSS). After completion of the RIE process, the residual SNS was removed with buffered oxide etch solution (10:1), and the surface reflectance was measured with an integrated-sphere-installed Cary-5000 UV-VIS/NIR spectrometer. For computational optical modeling, GD-Calc rigorous coupled-wave analysis (RCWA) software purchased from KJ Innovation was used.

## 3. Result and Discussion

### 3.1. Spin-Coating of 2-Dimensional SNSs

Prior to nanostructure fabrication, the uniform and high coverage of 2-dimensional (2-D) SNS (dia. = 310 nm) deposition must be realized in order to fabricate uniformly distributed high-density nanostructures. In this report, we used our novel solvent-controlled spin-coating method, which provides for the easy preparation of SNS solution followed by enhanced SNS coverage and uniformity on a large Si surface area. Unlike conventional SNS deposition techniques such as Langmuir–Blodgett (LB) dip-coating [29,30] and water-based spin-coating methods [24,31], our spin-coating process offers greatly reduced environmental sensitivity to vibration, humidity, and temperature in addition to the capability of expanded area SNS deposition. This is because DMF delivers more spin-coating-oriented solvent properties compared with conventional solvents (e.g., water), as shown in Table 1.

The DMF solvent-controlled SNS spin-coating method can provide three advantages compared with the conventional water-based spin-coating process. First, DMF delivers the reasonably low vapor pressure (VP) of 2.7 Torr (at 20 °C) compared with water (17.54 Torr at 20 °C), which leads to delayed SNS deposition and distribution time for DMF during spin-coating due to its slower evaporation rate. In the case of water, because of its high VP, it is essential to have humidity and temperature control or an additional surfactant mixture to delay the evaporation rate, as reported previously [24,32]. However, these previous approaches still have issues of low coverage or small area application. Second, the comparable viscosity of DMF to water prevents the discontinuous distribution of SNSs by maintaining a reasonably strong interaction between SNSs. Third, the improved wettability of DMF offers fast, uniform, and omnidirectional solution spread-out during coating as a result of reduced surface tension at the liquid/solid interface (i.e., low contact angle). The theoretical details of our DMF spin-coating method were reported in our previous study [33]. Because of these optimal solvent properties of DMF, we successfully spin-coated a 2-D SNS array on an Si surface with more than 95% monolayer coverage, as shown in Figure 1.

### 3.2. Nanostructure Fabrication with RIE Process

#### 3.2.1. SNS Size Reduction with CHF_3_/Ar Gases RIE

We obtained controlled nanostructure shapes by RIE fabrication after utilizing appropriate etching gas chemistries on the SNS-coated Si surface. The advantage of SNS lithography combined with RIE is the flexibility over the shape and scale of the nanostructure because of (1) the high etching selectivity between SiO_2_ (i.e., SNS) and Si, (2) the controllability of etching orientation by using various etching chemistries under different conditions, and (3) the ability to easily change the patterning scale by using different diameters of SNS [34]. In this work, for the flexibility of the nanostructure shape, a two-step RIE process was designed that utilized CHF_3_/Ar gases in the first step and Cl_2_ gas in the second step. To determine the etching duration for SNS size reduction, the SNS etching rate (ER) was investigated with four different etching times (50 s, 90 s, 150 s, and 180 s) on 310 nm diameter SNSs, and the results are shown in Figure 2. From the measured ER, we confirmed that highly linear SNS size reduction (0.72 nm/s) occurred until 150 s of etching, which most probably originated from the hard material nature of silica [34].

However, with 180 s of etching, a rapid increase of the ER was measured. From the SEM images in Figure 3, it can be seen that SNS size reduction occurred in both lateral and vertical directions, but vertical direction etching was more dominant and this originated from the anisotropic etching property with accelerated ion bombardment during RIE. Therefore, with excess etching times, the vertical thickness of SNS would become very thin, causing rapid ER acceleration in the lateral direction of SNSs.

#### 3.2.2. Si Vertical Etching with Cl_2_ Gas RIE

For vertical Si nanostructure (i.e., Si NP or NT) fabrication, three SNS mask sizes were used (272 nm, 201 nm, and 119 nm) and Cl_2_ gas etching was applied. From our Cl_2_ etching condition, a Si ER of ~ 150 nm/min was measured in the vertical direction. For this report, our target vertical height of Si NP was 1500 nm, so each sample had the same 10-min duration of Cl_2_ etching. The fabricated Si NPs are shown in Table 2 and the direct measurement results of each Si NP are listed. The top diameters of Si NPs were well defined with initial SNS diameters in the case of 272 nm and 201 nm having less than a 10% diameter mismatch, as shown in Table 2. However, in the case of the 119 nm initial SNS diameter, the top diameter of the fabricated Si NP did not follow the initial mask size.

This top diameter mismatch from the initial mask size can be explained by the (1) effective SNS mask area and (2) radical/ion flux ratio. Regarding the effective SNS mask area, as mentioned in Section 3.2.1, in SNS mask etching during CHF_3_/Ar RIE, the vertical-direction etching was more dominant than that in the lateral direction. As a consequence, the thickness of the mask became even thinner compared with the length of lateral mask size as etching time increased (Appendix A). Therefore, with longer etching times, the effective SNS mask area (orange solid circle in Appendix A) was reduced because little SNS remained at the edge region leads to less etching protection for the Si surface under the edge area of the mask compared with the core SNS region. The second aspect influencing the mismatch is the radical/ion flux ratio change with different mask sizes. Basically, RIE is an etching process that involves two etching species: (1) radical for chemical isotropic etching and (2) ion for physical anisotropic etching. With the Cl_2_ gas RIE process, the Cl_2_ molecules are dissociated into reactive neutral species, radicals, and positively charged particles (ions). The radicals react chemically with the Si surface, producing etching by-product (i.e., SiCl_x_), and ions collide with the Si surface and physically remove the by-product. However, as schematically shown in Figure 4, the radicals have a large angular distribution (AD) because of their neutral property, but ions move to the Si surface with a highly narrow AD because the charged ions are accelerated by applied bias [35]. Therefore, with a large SNS mask size (Figure 4a,b), ion-dominant vertical etching occurs and the lateral diameter of the Si NP is maintained with no significant changes (Figure 4d,e). However, with a small mask size, a higher radical/ion flux ratio will flow between the SNSs to etch the Si surface (Figure 4c), which offers more isotropic chemical etching followed by a reduced top diameter of the Si NPs from the initial mask size and resulting in Si NT fabrication (Figure 4f).

Unlike the top diameter, the control of the bottom diameter is a more complicated problem because it depends on both (1) the initial mask size and (2) the aspect ratio (height-to-NP diameter ratio) of the Si NPs. At the beginning of the etching, the bottom diameter is determined by mask size, but as etching time increases, the aspect-ratio-dependent etching (ARDE) phenomenon is more prevalent. This means that less radicals and ions will reach to the bottom of higher-aspect-ratio Si NPs, producing a reduced ER at the bottom (Appendix A). [36] Therefore, the bottom diameter will increase from the initial mask size to a pristine SNS diameter (i.e., 310 nm). In our report, the fabricated Si NPs and NTs had ~270 nm bottom diameter after 10-min Cl_2_ etching, which is similar to that of the pristine SNSs. This enlarged bottom diameter originates from the high-aspect-ratio structural features of our fabricated Si NPs and NTs.

#### 3.2.3. Optical Properties of Fabricated Si Nanostructures

In this section, we provide a theoretical and experimental study of our Si nanostructures for potential application in a photovoltaic (PV) device with enhanced light absorption properties. Note that for the theoretical study, three different top diameters (d_top_) of Si NPs were used (263 nm, 162 nm, and 20 nm) at the same 1.5 µm height, and a bottom diameter (d_bot_) of 270 nm was used in order to enable us to focus on the tapering effect. The controlled sharpening of NPs is crucial to the light harvesting for PV application because the tapered nanostructures produce a gradient refractive index change along the nanostructured layer, providing an excellent AR effect [37]. The effective refractive indices ($n_{eff}$) along the nanostructured layer were calculated as shown in Figure 5d–f by using the equation below [38]:$$n_{eff}={\left[f\cdot \;n_{Si}^{q}+\left(1-f\right)\right]}^{1/q},$$
where q is 2/3, $n_{Si}$ and $n_{air}$ are the refractive indices of the Si (3.62 @ λ = 900 nm) and air respectively, and f is a varied filling factor of Si NPs along with height. Based on the calculation, the Si NT structure produced a highly continuous and gradient refractive index change from top to bottom.

Unlike the NP structures (Figure 5a,b) that had abrupt refractive index changes in the layers (Figure 5d,e), the well-sharpened NT structure (Figure 5c) provides a gradient refractive index change along the Si NT layer, as schematically illustrated in Figure 6. This Si NT layer acts like multiply coated AR layers that can provide highly effective suppression of broadband surface reflection [39,40].

The results of measured surface reflection (Figure 7) confirm that the Si NT layer (d_top_ << d_bot_) had greatly reduced surface reflection compared with the other nanostructured layers (d_top_ = d_bot_ and d_top_ < d_bot_). By calculation of the weighted reflectance (R_w_) from the measured reflectance in the 300 nm to 1100 nm wavelength range, the Si NTs had only a 1.3% R_w_, which is a significantly reduced value from 5.7% (d_top_ < d_bot_) and 8.9% (d_top_ = d_bot_).

We also performed computational optical analysis with a rigorous coupled-wave analysis (RCWA) method, which is known as a highly accurate computational optical modeling technique for periodically arrayed nano-/micro-structures [41]. From our computational work, well-tapered Si NPs revealed greatly reduced surface reflection even with a large incident light angle for polarized light in both transverse electric (TE) and transverse magnetic (TM) modes. The computational analyses were performed with various incident angles (10°~89°) and 45° fixed azimuth angle at 600 nm wavelength for fabricated nanostructures (Figure 8). From the modeling, we found that decreased d_top_ provided less incident angle dependency for light reflection This is because, as shown in Figure 8a,b, larger diameter variation between d_top_ and d_bot_ reduced the reflection sensitivity for light in both the TE and TM modes owing to more effective refractive index change and highly randomized electron oscillation on the surface [7,42]. Consequently, the averaged reflection of TE and TM modes was effectively suppressed even at a large incident light angles, as shown in Figure 8c. Note that for RCWA modeling, the diffraction order was set up to 5th, but the reflectance was mostly made by 0th order because the period of the nanostructure was smaller than the used spectral range of 300~1100 nm.

From images (Figure 9) of our Si-NT-fabricated Si sample (2-inch dia. wafer, right side) and the plain Si surface (left side), it can be clearly seen that the Si-NT-structured Si surface produced barely any surface reflection and revealed the “black-Si” feature [43].

## 4. Conclusions

In this report, we introduced our novel solvent-controlled spin-coating method for highly uniform 2-D SNS layer deposition with enhanced coverage (>95%) on a large scaled surface area (2-inch diameter wafer). We showed that the combination of our SNS lithography technique with an RIE process could provide enhanced controllability for the fabrication of various nanopillar top diameter sizes. Experimental and computational modeling with RCWA proved that uniform nanostructures of high density with reduced d_top_ offered highly enhanced suppression of broadband surface reflection. In addition, reduced d_top_ also decreased the reflection sensitivity to incident light angle, which is a great advantage for producing more cost-effective solar modules without a light tracking system. Therefore, our nano-fabrication process has the potential to provide a great opportunity to apply a nano-fabrication process for low-cost and high efficiency Si PV fabrication.

## Figures and Tables

**Figure 1 micromachines-12-00119-f001:**
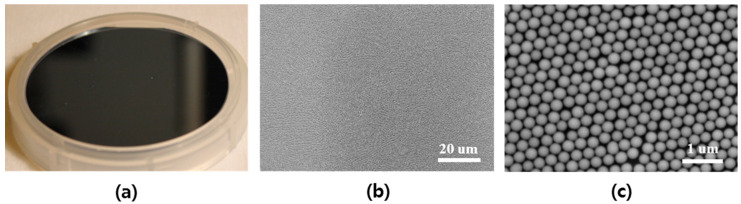
(**a**) Spin-coated 2-inch round Si surface, (**b**) SEM image of uniformly distributed Si nanospheres (SNSs), and (**c**) SEM image of closely-packed 2-dimensional (2-D) SNSs.

**Figure 2 micromachines-12-00119-f002:**
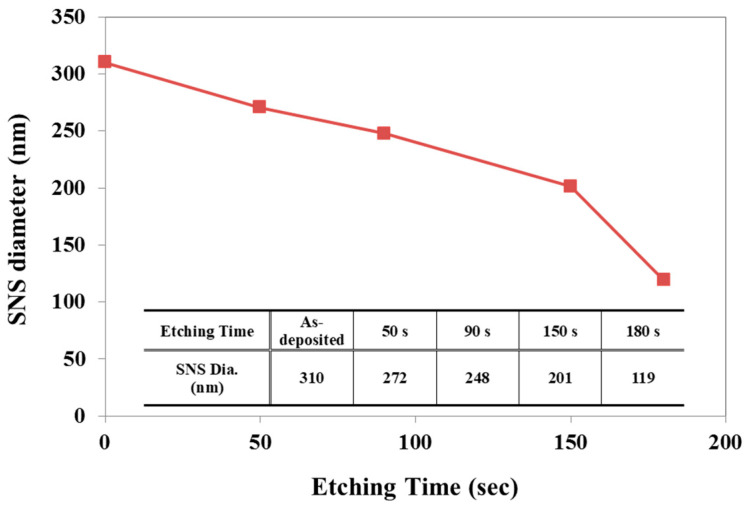
Measured SNS diameter with different CHF_3_/Ar RIE times.

**Figure 3 micromachines-12-00119-f003:**
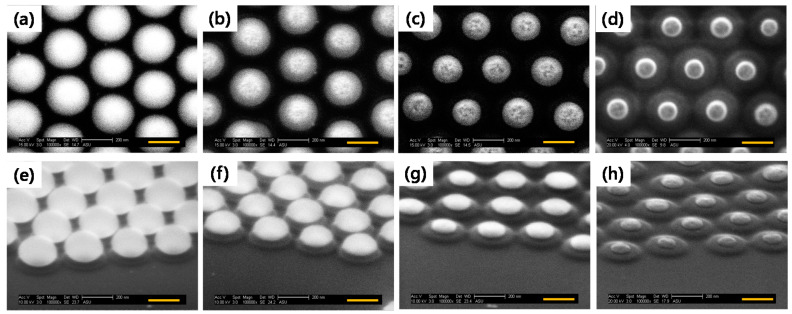
Top (**a**–**d**) and 25°-tilted (**e**–**h**) SEM images of SNSs after CHF_3_/Ar gases etching with four different etching times: (**a**,**e**) 50 s, (**b**,**f**) 90 s, (**c**,**g**) 150 s, and (**d**,**h**) 180 s.

**Figure 4 micromachines-12-00119-f004:**
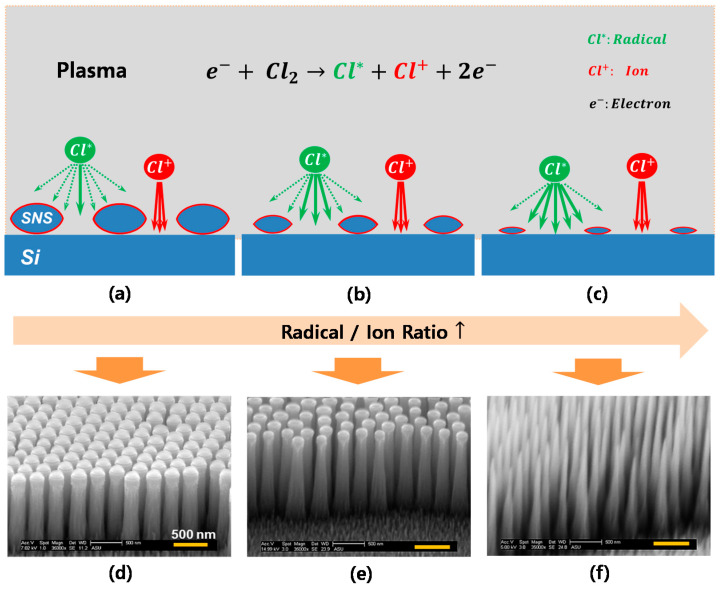
(**a**–**c**) Schematic illustration of radicals and ion angular distributions and radical/ion flux ratio change with SNS mask changes and (**d**–**f**) illustration of fabricated Si nanopillars (NPs) and nanotips (NTs) after 10-min Cl_2_ etching.

**Figure 5 micromachines-12-00119-f005:**
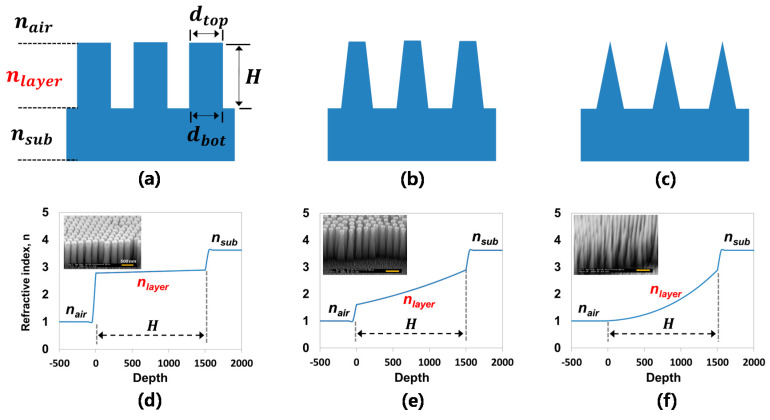
SEM images of (**a**) d_top_ = d_bottom_, (**b**) d_top_ < d_bottom_, and (**c**) d_top_ << d_bottom_ of nanostructures and (**d**–**f**) degree of refractive index change along with each nanostructured layer (Note: scale bar is 500 nm in SEM images).

**Figure 6 micromachines-12-00119-f006:**
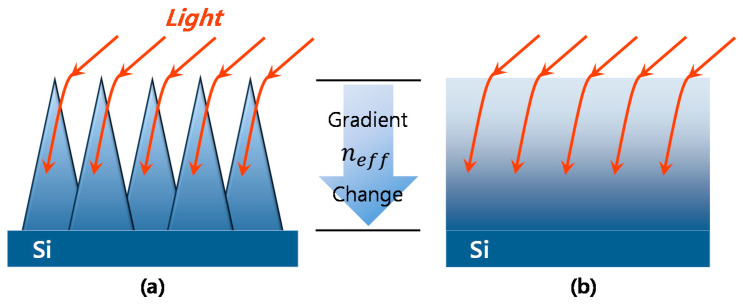
(**a**) Interaction of incident light with the Si NT array and (**b**) schematic illustration of the gradient refractive index change corresponding to (**a**).

**Figure 7 micromachines-12-00119-f007:**
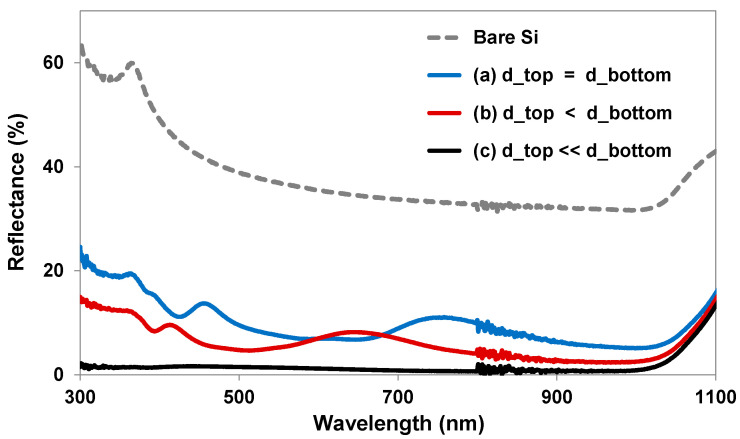
Measured reflectance of Si nanostructures with different top diameters.

**Figure 8 micromachines-12-00119-f008:**
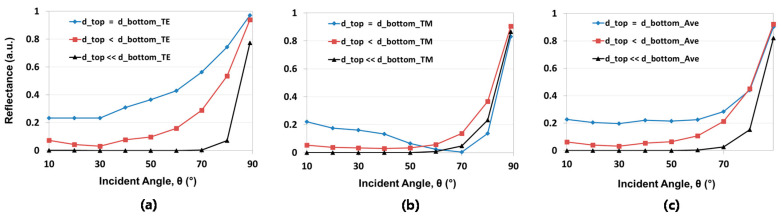
Calculated surface reflection at 600 nm wavelength with various incident angles, 10°~89°, and 45° fixed azimuth angle of light in (**a**) transverse electric (TE) mode, (**b**) transverse magnetic (TM) mode, and (**c**) averaged reflectance of TE and TM modes.

**Figure 9 micromachines-12-00119-f009:**
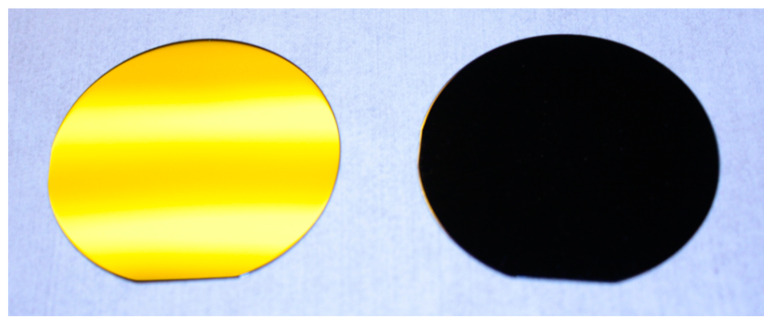
Surface images of 2-inch plain Si surface (**left**) and NT-structured Si surface (**right**).

**Table 1 micromachines-12-00119-t001:** Comparison of solvent properties between water and *N,N*-dimethyl-formamide (DMF).

	Solvent	Water	DMF
Properties			
Vapor Pressure (Torr, at 20 °C)		17.54	2.70
Viscosity (cP)		1.0	0.92
Wetting Angle (θ)		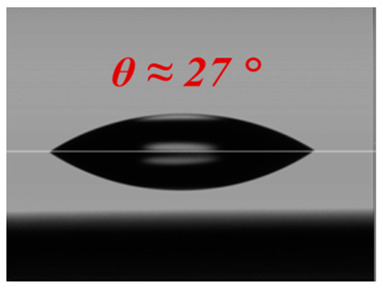	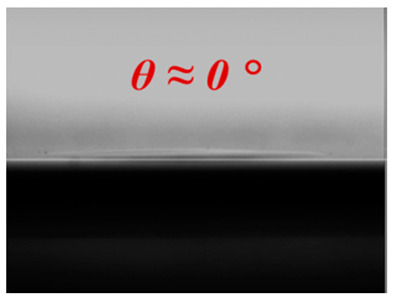

**Table 2 micromachines-12-00119-t002:** Measured top and bottom diameters and heights of fabricated Si NPs with different initial SNS mask size (unit: nm).

Initial SNS Dia.	272	201	119
SEM Images	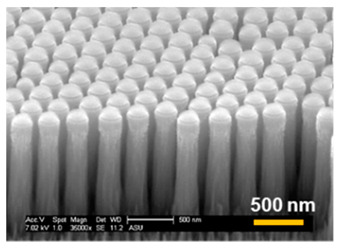	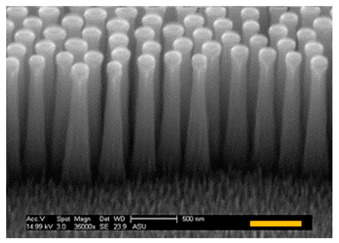	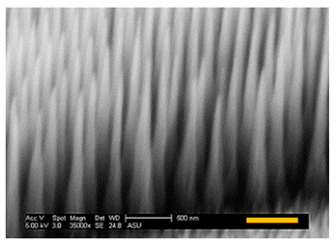
Top dia.	263 ± 12	162 ± 16	<20
Bottom dia.	281 ± 7	276 ± 8	269 ± 23
Height	1425 ± 22	1488 ± 27	1612 ± 38

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
