# Peer review of "Omnidirectional and Broadband Antireflection Effect with Tapered Silicon Nanostructures Fabricated with Low-Cost and Large-Area Capable Nanosphere Lithography"

_micromachines, 2021, doi:10.3390/mi12020119_

Round 1

Reviewer 1 Report

The authors present a study on the important problem of nanostructuring large surfaces of solar cells. They apply a previously developed method on nanoparticle lithography to this particular problem. The experimental details are very well detailed. The results are properly presented and described. In general, I find the manuscript interesting and relevant and therefore I support its publication. I have found a series of minor problems that I recommend the authors to fix.

Comments:

1. Standard deviations of diameters in Table 2 for the first two columns.

2. Line 8-211: I see problems in the sentence "the bottom diameter is a more complicated". Do the authors mean "The control of the bottom diameter is a more complicated problem"?

3. Line 8-224: The intermediate diameter 160 nm differs from the values of Table 2. Are the same three samples as in the table or dedicated samples for the optical characterization?

4. Line 9-243: The authors describe the arrangement of NP as a gradient index layer. Such an approach is only fully justified for the zero-th diffraction order, in general, and, in particular, in the subwavelength regime, i.e. when the lattice parameter is much smaller the wavelength. If the bottom radius is 270 nm and assuming that the distance between neighboring NPs is negligible, the lattice constant is approximately 500 nm. Hence, diffraction effects are expected for wavelengths larger than 500 nm. I would appreciate if the authors could comment on this, as Figure 7 does not show any features at 500 nm. Probably is related to comment 5.

5. In Figure 7, what are the conditions of the measurement, polarization, angle, specular only, or specular and diffuse?

6. What are the values of the filling factor for each sample?

7. Line 10-264: The sentence "We calculated ..." is difficult to read.

8. In line with comment 6. In RCWA, typically, the fields are expanded in diffraction orders. How many orders are considered in the results of Fig. 8? If only one has been considered, then no diffraction can be observed. In any case, I find the analysis fair enough for the discussion. I am only concerned with the clarity of the manuscript.

9. The address of the authors is not very accurate.

10. I could not locate "Bang Labs" company. The closer one I found is "Bangs Laboratories, Inc.". If it is the same, the authors could be more precise in reporting the name. It could be helpful to readers not so familiar with nanoparticle suppliers.

Author Response

Dear Reviewer,

I really appreciate your valuable comments to improve our manuscript. Here we provide author’s responses to your comments. 

Please refer to attached document.

Thank you .

Sincerely,

Reviewer 2 Report

The proposed method to make antireflective surface is not new. It is not correct statement that it is difficult to reach less than 10% reflectivity. solar cells are made using wet etching and with subsequent plasma etching the 0.1-1% reflectivity over visible spectral range can be easily obtained without any lithographic methods (https://doi.org/10.1016/j.solmat.2015.08.030; Anti-reflective surfaces: Cascading nano/microstructuring Y Nishijima et al. APL Photonics 1 (7), 076104 2016).

The formula for the effective refractive index is wrong. 

please show reflectivity in log-scale to clearly show what reflectivity is reached. it is not R=0.

Author Response

(The authors gave the same response as above.)

Round 2

Reviewer 2 Report

Some answers are provided, but some are ignored. 

The formula is cited by it does not make sense since for the f =  1 we do not get refractive index of Si but ((n_Si)^q)^q, really?

the scale lgR was not used It would show better what reflectivity was reached. I showed previously the papers where it was done. Such comparison would be useful since now it is not resolved in the plot.

Author Response

Dear Reviewer,

I really appreciate your valuable comments to improve our manuscript. Here we provide author’s responses to your comments.

Please refer to attached file.

Thank  you.

Sincerely,
